# The Nutritional Quality and Structural Analysis of Black Soldier Fly Larvae Flour before and after Defatting

**DOI:** 10.3390/insects13020168

**Published:** 2022-02-04

**Authors:** Bongisiwe Zozo, Merrill Margaret Wicht, Vusi Vincent Mshayisa, Jessy van Wyk

**Affiliations:** 1Department of Chemistry, Cape Peninsula University of Technology, Bellville 7535, South Africa; wichtm@cput.ac.za; 2Department of Food Science and Technology, Cape Peninsula University of Technology, Bellville 7535, South Africa; mshayisav@cput.ac.za (V.V.M.); vanwykj@cput.ac.za (J.v.W.)

**Keywords:** black soldier fly, insect, proximate, defatting, structural analysis, insect flour

## Abstract

**Simple Summary:**

The increasing global population and consumer demand for protein will render the provision of protein a serious future challenge, thus placing substantial pressure on the food industry to provide for the human population. The lower environmental impact of insect farming makes the consumption of insects such as black soldier fly larvae (BSFL) an appealing solution, although consumers in developed countries often respond to the idea of eating insects with aversion. One approach to adapt consumers to insects as part of their diet is through the application of making insect-based products in an unrecognized form. Nutritional value and structural properties of the BSFL flours (full fat and defatted) were assessed. Defatted as well as full-fat flour both displayed good nutritional and structural characteristics for use in many food applications.

**Abstract:**

This study aimed to assess the nutritional information and structural overview of the BSFL (black soldier fly larva) flours (full fat and defatted). The BSFL flours were obtained by freeze-drying the larvae and the removal of fat using hexane and isopropanol ratio of 3:2 (*v*/*v*), these solvents were used due to their defatting efficiency and because they are less toxic. Nutritional and structural analyses were conducted using standard methods. The full-fat and defatted flours had high protein content (45.82% and 56.11% respectively). Defatting significantly (*p* < 0.05) increases the protein content by approximately 10%, while the fat content decreased from 25.78% in full-fat larvae to 4.8% in defatted larvae. The compositional data were qualitatively confirmed with Universal Attenuated Total Reflectance Fourier Transform Infrared spectroscopy (UATR-FTIR) mainly in the amide I and II regions. Thermal gravimetry (TG) and differential scanning calorimeter (DSC) analysis, showed the conformational physical changes induced due to removal of fat which affected protein denaturation. DSC analysis displayed curves of both endothermic and exothermic reactions. During the first heating program, both samples had wide endothermic heating peaks ranging from 42 to 112 °C, which may be attributed to the water content in the samples evaporating. The first stage of the decomposition process was important, with loss of free and loosely bound water up to 150 °C, according to TGA curves. Protein and carbohydrates volatilized during the second stage of decomposition. The third level may be linked to polypeptide decomposition. FTIR revealed that the defatting process induced structural modifications on the amide I (1650 cm^−1^) and amide II (1540 cm^−1^) regions. Defatting has a significant effect on the functional groups and nutritional value of the BSFL. Defatted as well as full-fat flour both show good nutritional and structural characteristics for use in many food applications, however the improved proximate composition of the defatted BSFL can be applied to food products using BSFL flour as an alternative ingredient.

## 1. Introduction

Insects have long been a part of human diets, and they are still eaten by people in many parts of Africa [1]. They are thought to complement the diets of around 2 billion people [2]. Insects have attracted global attention as a possible alternative major source of proteins due to the current food shortage situation prevailing in many developing countries and future challenges of feeding over 9 billion people by 2050 [2]. The global food system is changing dramatically as a result of rising incomes, urbanization, environmental and nutritional problems, and other anthropogenic pressures. Because of these factors, there has been a significant shift in diets to include insect items, and this interest is likely to continue in the coming decades [3].

As a result, it is essential to assess alternative ingredients that are both inexpensive and readily available as replacements for traditional protein meals. In this sense, insects have drawn attention as a source of protein, amino acids (AA), fat, carbohydrates, vitamins, and trace elements [4]. The technique of removing fat from an insect is used to isolate high insect protein for animal feed from insect oil for feed and biofuel [5]. Defatted protein-rich meal and oil fractions from the insect have become commonplace in order to reduce biochemical compositional differences and the risk of lipid oxidation [6].

As previously mentioned, the nutrient composition of insects varies, with crude protein and crude lipid levels ranging from 40 to 54% and 15 to 49%, respectively, depending on the substrates fed to the larvae and the methods used to process them [5,7,8]. As a result, insect larvae can contain vital nutrients from sources that are not specifically appropriate for human or animal nutrition, a capability that can be used to tailor the insect larvae’s composition to desired nutrient profiles for use as feed ingredients.

Even with the great research interest in insects such as black soldier fly larvae (BSFL), data on nutrition information studies of defatted BSFL meals are still difficult to come by. Only a few studies have been conducted to date to assess the effects of defatted BSFL meal on a variety of aquatic animals [6,9,10]. The findings from these studies were inconclusive. This is most likely due to larvae reared on different feeds or the defatting methods used (e.g., mechanical press without solvents, Soxhlet process, and solvent extraction), which can make obtaining consistent quality insects’ meal and formulating diets with consistent nutrient composition difficult. For this study, solvent extraction was chosen, and a choice of solvents was made based on several other considerations include volatility (for easy removal later), the absence of harmful or reactive impurities (to prevent fat reactions), the ability to form a two-phase system with water (to remove non-lipids), and price. In this study, a hexane:isopropanol blend was chosen. According to Schaumburg and Spencer (1978) [11], hexane is a known neurotoxin that is converted to 2,5-diketo compounds in high concentrations, but it is relatively non-toxic in laboratory use and does not pose a threat after sewage treatment. While many solvents have been recommended [12,13], the chloroform:methanol system of Folch et al. (1957) [14] and its various modifications are the most common. In this regard, chloroform has been shown to cause tumors in animals, and methanol is known to cause vision damage.

Recently, research on different substrates affecting the composition of BSFL has been well reported [15,16,17]. However, the nutrient quality and structure overview of both whole (full fat) and defatted BSFL have not been studied as a suitable alternative protein source. This research aimed to assess the effect of defatting on the BSFL nutrient quality, mineral composition, and structural analysis. In addition, to understand the mechanistic speculation of increased protein yield in this insect species, the compositions and structural variation of the insect product were investigated using DSC and TGA.

## 2. Materials and Methods

### 2.1. Chemicals

All chemicals used in the study were of analytical grade unless otherwise stated. Hexane (Analytical reagent grade), isopropanol (88%), petroleum ether (Analytical reagent grade), HNO_3_ (55%), Ethylenediaminetetraacetic acid (EDTA, Certified Reference Material, 41% Carbon, 5.5% Hydrogen, 9.56% Nitrogen), and potassium hydroxide were obtained from Sigma-Aldrich (Johannesburg, South Africa). Stock standard solution (1000 ppm) of the minerals was also obtained from Sigma-Aldrich (Johannesburg, South Africa). Prepared solutions and reagents were stored under conditions that prevented deterioration or contamination. All solutions were prepared using deionized water (Milli-Q system, Millipore, Gradient model) with a resistivity of 18.2 MΩ·cm.

### 2.2. Sample Processing

Black soldier flies in the larval stage (after 14 days) were supplied by AgriProtein, Phillipi, Cape Town, South Africa. The supplied BSFL was fed clean waste streams such as vegetables and organic waste. They were immediately cleaned and blanched at 85 °C for 3 min to prevent browning and were stored at −80 °C until further processing. The frozen insects were ground in a laboratory blender (Bamix, Checkers, Cape Town, South Africa) and then freeze-dried (VirTis SP Scientific, wizard 2.0, Johannesburg, South Africa) for 3 days at 500 mTorr; freeze-drying was used to obtain a more stable and easier product to use.

Some of the freeze-dried samples were subjected to a defatting process. The insect was defatted using hexane and isopropanol at a ratio of 3:2 (*v*/*v*). One part of the whole ground BSFL sample (g) and five parts of the solvent mixture (mL) were stirred on a magnetic stirrer (Labotec, Cape Town, South Africa) for 2 h. Following sedimentation of the solids, the solvent–fat mixture was decanted. The procedure was repeated twice. Residual solvent was removed by evaporation in the fume hood overnight. Subsequent fine grinding of the defatted samples using mortar and pestle and sieving through a stainless-steel filter sieve with a pore size of 500 μm to remove the integument, produced defatted BSFL flour.

### 2.3. Proximate Analysis

Proximate analysis, including protein content, moisture content, ash content, and crude fat was performed according to the standard methods proposed by the Association of Official Analytical Chemists (AOAC). Carbohydrates were calculated by difference: carbohydrates = 100 − (% proteins + % lipids + % ash) according to the method described by Pearson’s Chemical Analysis of Foods.

#### 2.3.1. Moisture Content

Before analysis, whole and defatted insects were pulverized separately with a mortar and pestle. The moisture content of the samples was measured by drying them at 100 °C in a drying oven (Scientific series 900). A crucible was removed from a hot oven (100 °C) and put in a desiccator for 30 min to cool. The crucible was precisely measured using an analytical balance (Chem-Lab Supplies, Cape Town, South Africa). The scale was calibrated, and 5 g of sample were accurately weighed into the crucible (61 × 37 mm porcelain, Chem-Lab Supplies, Cape Town, South Africa). The crucible and sample were put in the oven for 12 h. The crucibles were taken out of the oven after drying and put in a desiccator to cool for 30 min. Weighing the moisture-free samples for the purposes of calculation, all the masses were reported using Equation (1).

Calculation:(1)$$\mathrm{Moisture}\;\left(\%\right)=\frac{\left(A+B-C\right)}{B}\times 100$$
A—crucible weight;B—sample weight;C—weight of the crucible and sample after completion of the oven-drying procedure.

#### 2.3.2. Ash Content

Whole and defatted insects were ground separately using a mortar and pestle or a countertop blender (Philips, Eindhoven, The Netherlands) until a fine powder was obtained before analysis. After ashing the samples in a muffle furnace (Carbolite Sheffield LMF 4) at 500 °C for 12 h, the ash content was determined. The sample was first dried using the previously mentioned method for determining moisture content. The sample was then moved to the furnace. Afterward, the furnace was turned off and the sample was allowed to cool for at least 2 h before being placed in a desiccator for 30 min, followed by accurate weighing. Masses were recorded for further calculations using Equation (2).

Calculation:(2)$$\mathrm{Ash}\;\left(\%\right)=\frac{D-A}{\mathrm{sample}\;\mathrm{weight}}\times 100$$
A—weight of the clean crucible;D—weight of the crucible and ash.

#### 2.3.3. Protein Content

Crude protein content was determined using Dumas (TruSpec^TM^ Leco Carbon/Hydrogen/Nitrogen Series) which was calibrated with EDTA (AOAC, 2000), using a protein-to-nitrogen conversion factor of 5.60 [18]. A small piece of tin foil cup (Leco Corporation 502167, St. Joseph, MI, USA) was placed on the scale. The scale was tared, and the foil piece was filled with 0.1 g of the sample. The scale’s sample-filled foil was removed and sealed into a teardrop shape. The teardrop-shaped foil was mounted in the TruSpecTM Leco’s carousel. The sample was then analyzed, and a nitrogen (N) value was shown on the computer screen. The N value was recorded for further calculation using Equation (3).

Calculation:
Crude protein (%) = Nitrogen value % × 5.60(3)

#### 2.3.4. Fat Content

The Soxhlet process, as defined by the Association of German Agricultural Investigation and Research Institutions (VDLUFA, 1976), was used to determine fat content, with some modifications. The 3-g homogenized samples were placed in extraction thimbles (26 × 60, Bio-Smart Scientific, Johannesburg, South Africa). The thimbles were placed into the Soxhlet compartment (ST 243 Soxhlet 1040 extraction unit). The extraction was implemented using 50 mL of petroleum ether (bp. 40–60 °C) in the extraction cups for 30 min at 60 °C in the Soxhlet apparatus. After extraction, the petroleum ether was evaporated using a heating block at 50 °C under a fume hood using extraction cups. The fat residue was cooled in a desiccator for 30 min and weighed. Masses were recorded for further calculations using Equation (4).

Calculation:(4)$$\mathrm{Fat}\;\left(\%\right)=\frac{\left(\mathrm{crucible}+\mathrm{oil}\right)-\mathrm{empty}\;\mathrm{crucible}}{\mathrm{sample}\;\mathrm{weight}}\times 100$$

### 2.4. Mineral Composition

The mineral analysis was carried out at the Cape Peninsula University of Technology, Chemistry Department at Bellville Campus. The mineral content of 5 g of dried and finely ground samples was determined. Each sample was ashed at 500 °C for 5 h (constant weight was reached) and left to cool to ambient temperature. Afterward, 5 mL of 1 M HNO_3_ was added. Subsequently, this was filtered into a 100-mL volumetric flask, which was made up to volume with 1 M HNO_3_. Elements were measured with a vertical quartz torch and a Cetac ASX-520 autosampler on an iCAP 6000 Series Inductively Coupled Plasma (ICP) Spectrophotometer (Thermo Electron Corporation, Strada Rivoltana, 20090 Rodana, Milan, Italy). The iTEVA Analyst program was used to measure the element concentrations. The camera temperature was −27 °C, the generator was 24 °C, the optics were 38 °C, the RF power was 1150 W, the pump rate was 50 rpm, the aux gas flow was 0.5 L/min, the nebulizer was 0.7 L/min, the coolant gas was 12 L/min, and the usual purge gas flow was 2–5 mL/min. Wavelengths for the elements were as follows: Ca 396.847 nm, Fe 259.941 nm, K 766.491 nm, Mg 279.553 nm, Mn 257.611 nm, Na 588.995 nm, and Zn 213.859 nm. Working standards of 0.1, 0.2, 0.4, 1.0, 2.0, and 3.0 ppm were prepared in 0.5 M HNO_3_.

### 2.5. Thermal and Structural Analysis

#### 2.5.1. Thermal Gravimetric Analysis (TGA)

A thermogravimetric e was used to assess the thermal degradation properties (Perkin Elmer, TGA 7, Waltham, MA, USA). Each sample was weighed and put on the heating pan at a concentration of about 10 mg. To prevent thermo-oxidative degradation, the experiments were carried out under nitrogen atmosphere, ramp mode with a temperature range of 30 to 600 °C and a steady heating rate of 10 °C min^−1^ (20 mL min^−1^).

#### 2.5.2. Differential Scanning Calorimetry (DSC)

The thermal test was carried out using an ASTM D3418-compliant differential scanning calorimeter (DSC822e, Mettler Toledo, Greifensee, Switzerland). Under a nitrogen atmosphere, in a semi-hermetic tray, approximately 10 mg of the samples were heated at a rate of 10 °C min^−1^ from room temperature to 200 °C. The samples were cooled to 30 °C and kept for 5 min, and then heated to 400 °C at a rate of 10 °C min^−1^ after a 5 min holding period to eliminate the thermal background.

#### 2.5.3. Universal Attenuated Total Reflectance Fourier-Transform Infrared Spectroscopy (UATR-FTIR)

All samples were analyzed using a Perkin Elmer FTIR equipped with a UATR polarization accessory (Thermo Electron, Waltham, MA, USA). All spectra were obtained by combining 32 scans in the range of 4000–400 cm^−1^ at a resolution of 4 cm^−1^. A context spectrum was collected before each sample’s data was collected. To extract any residual contribution from previous samples, the UATR crystal was washed with acetone. Sample powders obtained by grinding in a mortar and pestle were placed directly covering the surface of the ATR crystal for the FTIR analysis Data analysis.

All data were subjected to multivariate analysis of variance (MAN-OVA) to ascertain whether the main effects resulted in significant differences in response variables. Duncan’s multiple comparison post hoc test was used to test significant differences (*p* < 0.05) between individual means. SPSS 27.0 for Windows^®^ was used for the statistical analyses and the level of confidence required for significance was selected at *p* < 0.05.

## 3. Results

### 3.1. Proximate Composition of Full-Fat and Defatted BSFL Flours

The composition of the BSF larvae flour is summarized in Table 1. Given the outcomes of the proximate analysis, the numerical comparisons in the topic and conclusion parts of these findings were statistically analyzed and all analyses were done in triplicate (n = 3). These results and those published were compared, and it should be remembered that the crude protein content of the entire BSFL ranged from 30.6% to 43.6% [19,20] in literature.

This study’s crude protein content (45.8%) was comparable to that recorded by St-Hilaire et al. (2007) [20], this may be because the larvae were raised on the same feed substrate (clean feed), collected at the same time (larvae stage), and handled in the same way (freeze-drying). The small differences in the day of harvest within the larval stage may explain the difference; for example, Aniebo and Owen (2010) [21], found that crude protein content decreased from 59.6% to 54.2% to 50.8% in oven-dried BSF larvae harvested on different days between 14–18 days of the larval stage, for the first, second, and third days, respectively. When compared to Pieterse’s (2014) [19] 30.6% crude protein content, the current study’s crude protein content was substantially higher.

The moisture in BSF from the literature [19,22] revealed levels in the region of 0.73% lower than the results observed from the whole BSFL in this study (Table 1). As the fat content of the whole larvae decreased, there was an increase in protein content. Defatted BSFL is equal to the whole BSFL minus the fat removed (100 − 25.78 = 74.22 g). Therefore, after defatting the proximate analyses values should theoretically be similar to the experimental values. Even though the theoretical proximate values are based on the initial experimental values of the whole BSFL, the assayed values of the defatted BSFL are well correlated. The values (experimental (theoretical)) are as follows: moisture 6.46 (5.59); protein 56.11 (61.74); ash 8.08 (9.23); and CHO 24.49 (23.45).

### 3.2. Mineral Composition

Since the ash content of a sample reflects the minerals it contains, BSFL flours are therefore very rich in minerals as shown in Table 2. Many minerals can be found in the BLSF flours, major elements such as sodium (Na), calcium (Ca), magnesium (Mg), potassium (K) and the trace elements such as iron (Fe), manganese (Mn) and zinc (Zn). Calcium levels in BSF flours were low from our experiment when compared to the levels reported by [5]. The content of Ca and iron (Fe) seems to be unaffected by the removal of fat. The removal of fat raised the content of Mg, Mn, K, and Zn (53.56 to 509.08 mg/Kg, 20.03 to 54.79 mg/Kg, 254.60 to 509.73 mg/Kg, and 0.24 to 80.77 mg/Kg, respectively); however, Na content was decreased by the extraction of fat from 354.72 to 47.56 mg/Kg. It could be that some minerals are bound to the fat component. Unfortunately, there is no evidence or results that support the previous statement as definite. The results show that Mg and K have the highest content while Fe recorded the lowest content on the defatted sample. These results are comparable with the findings of Mg (530 mg/Kg) and K (594 mg/Kg) of black soldier fly larvae prepupae reported by Spranghers et al. (2017) [23]. Other elements except Ca, may show differences based on the type of feed and time harvest. Copper (Cu) and argon (Ar) were not detected in both the whole and defatted BSFL flours.

### 3.3. Thermal and Structural Analysis

TGA curves of the whole and defatted BSFL are shown in Figure 1. It was discovered that the first stage of the decomposition process was significant, as it resulted in the loss of free and loosely bound water up to 150 °C. Protein and carbohydrates volatilized during the second stage of decomposition, which took place between 150 and 420 °C. The third level, which occurs between 420 and 550 °C, may be linked to protein decomposition. The whole and defatted BSFL samples remained with the weight of 25% and 27%, respectively, this is due to polymers that decompose at very high temperatures. Further denaturation and unfolding of the BSFL protein structure occurred at higher temperatures for both samples, therefore, they are thermally stable.

The DSC profile of BSF larvae whole and BSF larvae defatted are shown in Figure 2. It was observed that the two samples showed endothermic peaks and the highest enthalpy changes occurred at temperatures between 80 and 110 °C. Table 3 displays the thermal profile of the whole and defatted (DF) samples.

This is observed since the whole sample is deemed to be more energy-dense than defatted samples. Thus, the whole sample needs more energy to promote polypeptide decomposition. This phenomenon is in accord with TGA analysis observed in Figure 1. A higher temperature was required for denaturation (TGA) and more energy was required for decomposition (DSC).

Figure 3 showed the FTIR spectra of the whole and defatted BSFL samples under the optimum extraction condition. Diagrams revealed that the regions of 1650 and 1540 cm^−1^ which referred to C=O and C–N stretching from amides I and II were modified by the defatting because the intensity of the whole BSFL decreased compared to the DF BSFL sample. The functional group region at 2930 cm^−1^ referred to as sp^2^, sp^3^ C–H as the results of fat showed less intense peaks for the DF BSFL. As shown in Figure 3, the aliphatic C–H stretching vibration had a high absorbance between 2853.43 and 2923.30 cm^−1^, showing a considerable amount of methyl and methylene groups. The CO group stretching vibration in ketones or carboxylic acids was defined by the absorption peak around 1742.86 cm^−1^, which was consistent with the high content of triglycerides which contain ester functions in the lipid.

At 1574.38 cm^−1^, peaks were found that corresponded to the involvement of alkenes (–C=C– stretch). The triglyceride functional groups were found in insect flours, which was confirmed by these peaks. The 1165.75 cm^−1^ absorption band indicated the existence of aromatic amine (C–N stretch). At 1095.45 cm^−1^, the ester group (C–O stretch or C–H bend) was discovered. The extraction of lipid from the yellow mealworm beetle yielded similar findings previously [25,26,27].

## 4. Discussion

Prior work has documented the importance of insect consumption, insects such as BSF, to meet the pressure of imminent food scarcity. As an alternative protein source, insect meals have been reported to have various beneficial effects in both production and health of animals, for example in the animal nutrition industry it has become a necessity to seek sustainable and alternative protein sources for animal production. However, these studies have either been short-term studies or have not focused on the exploitation of the BSF insect for human consumption. The experimental part of this study was the extraction of the oil (by-product) from BSFL using a hexane:isopropanol mixture. This solvent mixture successfully removed most of the fat only leaving behind 4.86%. The disparity in protein values of 5.63 can be explained by the fact that fat was not completely eliminated. The defatted BSFL also showed a higher moisture content than the whole BSFL, it would therefore indicate that there was moisture absorption during the storage of the flour. It was discovered that the whole larvae and defatted larvae both have high nutritional value, with 45.82% and 56.11% protein content, respectively. The authors’ findings of high crude protein values may be due to differences in the larval growth medium, harvest point, and processing system. This distinction can also be explained by the presence of a chitin layer that contains nitrogen–hydrogen bonds on the prepupae [28].

The effect of removing fat from the BSF larvae was further evaluated. Nutritional information was determined, and further evaluation was completed by structural and thermal instrumentation. The FTIR, TGA, and DSC instrumentation were used to confirm structural differences and changes among the full fat and defatted BSFL flours. FTIR was specifically used to identify some of the functional groups present and detail conformational modifications that occur when fat is removed. As for the essential mineral, the human body has a specific amount required, for example, calcium which is for building strong bones, the reported results fell within the recommended dietary allowance [24], providing a considerable advantage to BSFL over other food sources, nutritionally.

FTIR confirmed that the decrease of fat leads to an increase in the structural change of the protein. At the same time, the spectroscopic analysis of FTIR indicated that amide I and II were modified by the removal of fat. The shifts of amide bonds in the FTIR analysis and the DSC peak shifts of the samples further testify to the removal of the fat. The results of this study also demonstrate that the thermal stability and nutritional value of the flours were effectively improved by the removal of fat. Therefore, the present results suggest that defatted BSF flours can be incorporated into existing food products.

## 5. Conclusions

Both the whole and defatted larvae were found to be significantly high in crude protein, suggesting that they could be investigated further for protein digestibility, protein-rich concentrate, or feed ingredient in ruminant feeding. Different strategies for isolating or draining the oil completely from the samples should be tested in the future. It could be inferred that black soldier fly flour can be successfully utilized as an alternative protein source that can partially substitute other protein sources while still allowing regular diets to be maintained. However, it is recommended that, in addition to the removal of fat and oil, the residual flour should be processed for protein extraction and characterization thereof. Additionally, both BSFL flours are high in trace minerals. Finally, the defatting treatments resulted in higher crude protein values than the full-fat treatment. As shown in the DSC and TGA analysis, the results of this study demonstrate that the thermal stability of the flours was effectively improved by the removal of fat. The effect of defatting yields a higher-level protein for further development for animal consumption and may be considered to be incorporated in human food.

## Figures and Tables

**Figure 1 insects-13-00168-f001:**
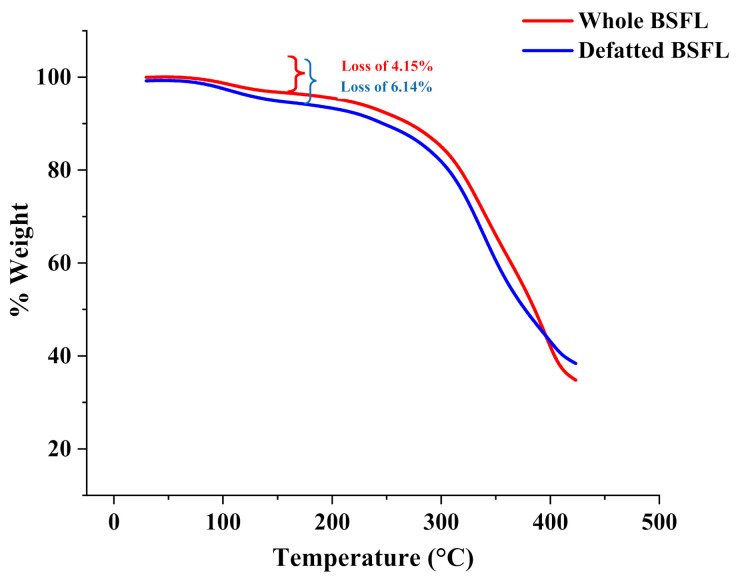
The thermal gravimetric curves of the whole and defatted black soldier fly larvae.

**Figure 2 insects-13-00168-f002:**
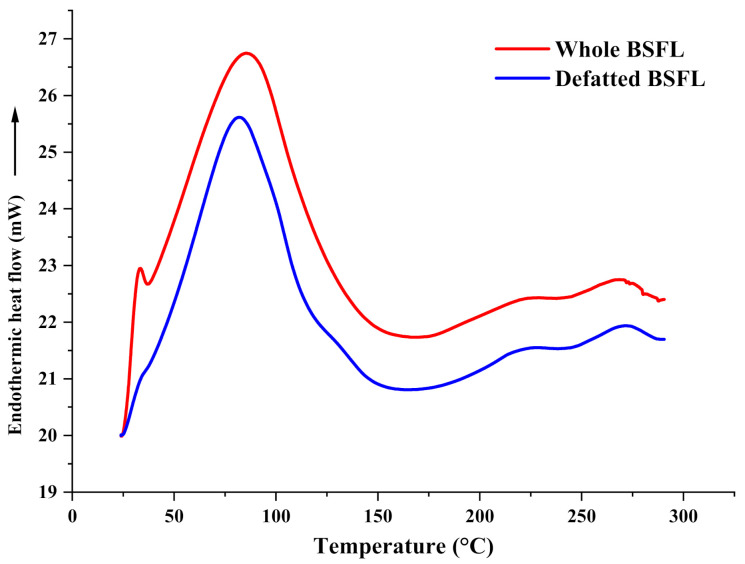
The differential scanning calorimetry profile of the black soldier fly larvae whole and black soldier fly larvae defatted.

**Figure 3 insects-13-00168-f003:**
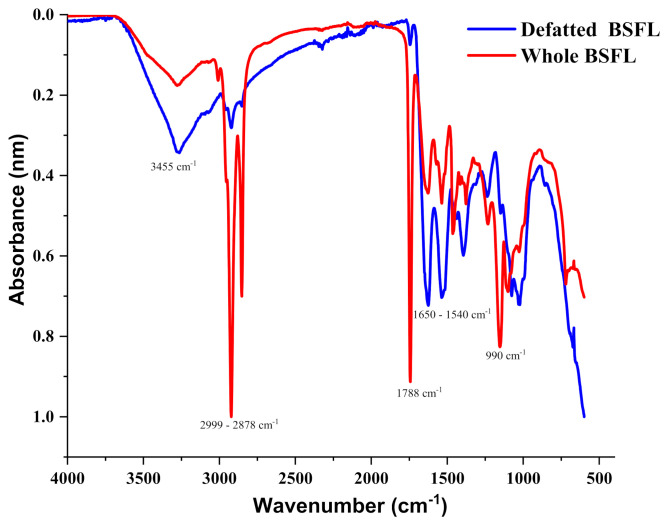
The Fourier transform infrared spectroscopy spectra of the whole and defatted black soldier fly larvae.

**Table 1 insects-13-00168-t001:** Proximate analyses (dry-matter basis) of the whole BSFL and defatted BSFL flour (DF BSFL). Values represent mean ± standard deviation (n = 3).

Treatment	Moisture %	Protein %	Fat %	Ash %	CHO * %
Whole BSFL	4.14 ± 0.05	45.82 ± 0.14	25.78 ± 1.67	6.85 ± 0.34	17.41 ± 0.03
DF BSFL	6.46 ± 0.02	56.11 ± 0.53	4.86 ± 0.06	11.39 ± 0.14	21.19 ± 0.56

* CHO (carbohydrate chemical abbreviation).

**Table 2 insects-13-00168-t002:** Mineral composition of the whole BSF, defatted (DF) BSF larvae flour, and recommended dietary allowance [24] ^1^.

Minerals	BSFL		Recommended Dietary Allowance (mg per Day)
	Whole BSFL (mg/Kg)	Defatted BSFL (mg/Kg)	
Ar	not detected	not detected	-
Ca	20.31 ± 0.11	20.58 ± 0.02	600
Cu	not detected	not detected	-
Fe	0.249 ± 0.10	0.975 ± 0.03	17
Mg	53.56 ± 0.04	509.08 ± 0.02	310
Mn	20.03 ± 0.01	54.79 ± 0.02	40.10
K	254.60 ± 0.01	509.73± 0.01	3225
Na	354.72 ± 0.25	47.56 ± 0.02	1902
Zn	0.24 ± 0.01	80.72 ± 0.01	12

^1^ Mean values ± standard deviation (n = 3).

**Table 3 insects-13-00168-t003:** Thermal profile of the whole and defatted (DF) samples from DSC analysis (TO, onset temperature; TP, peak temperature, and ΔH (peak enthalpy)). Duplicate samples were analyzed and the results were presented as mean values.

	T_0_ (°C)	T_P_ (°C)	ΔH (Jg^−1^)
Whole BSFL	40.76	86.68	197.29
DF BSFL	40.76	81.74	153.98

## Data Availability

The data presented in this study are available in article.

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
