# Peer review of "The Nutritional Quality and Structural Analysis of Black Soldier Fly Larvae Flour before and after Defatting"

_insects, 2022, doi:10.3390/insects13020168_

Round 1

Reviewer 1 Report

Dear authors:

The article has a timely and interesting topic, and I think it is appropriate for this journal. However I think you still need to have in account some issues on their manuscript:

  1. I understand it is a communication. A communication should be shorter. This manuscript having a improved discussion section and probably to include more samples could be an interesting original research paper.
  2. The results and discussion sections should be separate. The authors present interesting results but the discussion of these results is scarce and does not clearly reflect the applications these results could have.
  3. Tables 3.2 and 3.3 lack information about their content, samples number, etc. Tables should be clear and understood without having to resort to the article´s text.
  4. The number of samples is not clear and whether they were analysed in duo.
  5. Thermal and structural analyses are interesting, however it lacks a proper discussion how they represent advantages or disadvantages in the food industry. They say in the abstract they found structural modifications, but in results it is not clear and how it could be good for human food or as a food ingredient. Indeed, I miss some discussion regarding protein digestibility, what happen to chitin content?
  6. Since there is not a clear discussion, conclusions are not connected to results.

Reviewer 2 Report

Author did good effort in the nutritional quality and structural analysis of Black soldier 2fly larvae flour before and after defatting, but I have some questions need author to answer

  1. In abstract, Expression “ while the fat content decreased from 8% in full-fat larvae to 1.8% in defatted larvae. It means 8% fat in full-fat larvae and 1.8% fat in defatted larvae. But in text Table 3.1, 25.78 ± 1.67% fat was in full-fat larvae(Whole BSFL), 4.86 ± 0.06% fat was in in defatted larvae(DF BSFL). Which is right?
  2. Author should introduce the background of black soldier fly larvae, such as the diet of BSFL.
  3. In Results and Discussion

“3. Results and discussion

3.2. Proximate composition of full-fat and defatted BSFL flours”

I do not see 3.1 part

  1. The writing is not good. Author should ask local English expert to edit language.
  2. Author should explain why you do thermal and structural analysis.
  3. Author should explain how is the relationship between nutritional, structural characteristics of both Defatted as well as full-fat flour with food applications.

Reviewer 3 Report

Paper is interesting and has average scientific value, therefore it is suitable for publication.

Round 2

Reviewer 2 Report

Author answered and revised my questions and comments one by one. It meet my requirements. I have no more comments!